# Enhanced Site-Specific Fluorescent Labeling of Membrane Proteins Using Native Nanodiscs

**DOI:** 10.3390/biom15020254

**Published:** 2025-02-10

**Authors:** Bence Ezsias, Felix Wolkenstein, Nikolaus Goessweiner-Mohr, Rohit Yadav, Christine Siligan, Sandra Posch, Andreas Horner, Carolyn Vargas, Sandro Keller, Peter Pohl

**Affiliations:** 1Institute of Biophysics, Johannes Kepler University Linz, Gruberstraße 40, 4020 Linz, Austria; bence.ezsias@jku.at (B.E.); nikolaus.goessweiner-mohr@jku.at (N.G.-M.); rohit.yadav@jku.at (R.Y.); christine.siligan@jku.at (C.S.); sandra.posch@jku.at (S.P.); andreas.horner@jku.at (A.H.); 2Biophysics, Institute of Molecular Biosciences (IMB), NAWI Graz, University of Graz, Humboldtstr. 50/III, 8010 Graz, Austria; carolyn.vargas@uni-graz.at (C.V.);; 3Field of Excellence BioHealth, University of Graz, 8010 Graz, Austria; 4BioTechMed-Graz, 8010 Graz, Austria

**Keywords:** potassium channel, urea channel, electrophysiology, fluorescence correlation spectroscopy

## Abstract

Fluorescent labeling of membrane proteins is essential for exploring their functions, signaling pathways, interaction partners, and structural dynamics. Organic fluorophores are commonly used for this purpose due to their favorable photophysical properties and photostability. However, a persistent challenge is the inaccessibility of the surface-exposed cysteine residues required for site-specific labeling, as these residues often become sequestered within detergent micelles during protein extraction. To address this limitation, we developed an approach based on polymer-encapsulated nanodiscs that preserves the protein’s native-like lipid-bilayer environment while ensuring the accessibility of surface-exposed cysteine residues. In this method, His-tagged proteins embedded in native nanodiscs are retained on a nickel affinity column, allowing for simultaneous purification and labeling by adding fluorescent dyes. This versatile technique was demonstrated with two challenging-to-label membrane proteins, the potassium channel KvAP and the urea channel *Hp*UreI, for which detergent-based labeling had failed. This opens new possibilities for studying a wide range of fluorescently labeled membrane proteins in near-native states, advancing applications in biophysics, structural biology, and drug discovery.

## 1. Introduction

Fluorescent protein labeling enables the real-time, high-resolution monitoring of protein dynamics, interactions, and functions in both live cells [1,2,3] and reconstituted systems [4,5]. While proteins can exhibit intrinsic fluorescence, extrinsic labeling offers distinct advantages. Among the most common labeling methods are genetically encoded fluorescent proteins and organic fluorophores.

Unlike genetically encoded fluorescent proteins, organic fluorophores offer benefits such as smaller sizes, broader spectral options, higher quantum yields, and site-specific labeling [6]. These dyes can be conjugated to reactive groups on protein surfaces, including amino acid residues with surface-exposed side chains [7] or artificially introduced tags [8,9]. Despite the benefits, achieving efficient and specific labeling remains challenging, as reactions can be limited by the accessibility of targeted residues. The efficient removal of excess dye is also critical and is typically achieved through affinity or size-exclusion chromatography (SEC).

Among potential labeling targets, cysteine residues are particularly attractive due to their reactivity and low abundance in proteins [10]. Thiol-reactive compounds, such as maleimides, can form stable covalent bonds with cysteine residues under mild conditions through a Michael-type addition reaction [11,12], provided that the cysteine’s thiol group is in a reduced state maintained by reducing agents like dithiothreitol (DTT) or tris(2-carboxyethyl)phosphine (TCEP). This reaction offers a selective and efficient labeling method suitable for mild, near-physiological conditions [13].

For in vitro studies, membrane proteins are often extracted from cells and reconstituted into lipid vesicles [14]. Detergent solubilization is a standard approach for extracting membrane proteins, where detergent molecules disrupt the native lipid bilayer and surround the extracted protein to form protein/detergent micelles [15]. However, detergents often obscure surface-exposed cysteine residues, rendering them inaccessible for fluorescent labeling. We encountered this issue with challenging-to-label proteins such as the archaeal voltage-gated potassium channel KvAP from the thermophilic Archaea *Aeropyrum pernix* [16] and the *Helicobacter pylori* urea channel *Hp*UreI [17]. KvAP consists of four subunits, each containing six transmembrane segments (S1–S6). The S1–S4 segments form the voltage-sensing domain, while the S5 and S6 segments comprise the pore domain (Figure 1A). The pore facilitates the conduction of dehydrated K^+^ ions through a conserved glycine–tyrosine–glycine motif in the selectivity filter [18], ensuring the exclusion of other ions [19,20]. *Hp*UreI, a proton-gated urea channel, enables the rapid uptake of urea. Its enzymatic degradation by an attached cytoplasmic urease helps maintain a neutral pH, thereby contributing to bacterial survival in the acidic conditions of the stomach [21]. Structurally, *Hp*UreI forms a hexamer, with each monomer comprising six transmembrane helices that together form a central pore (Figure 1B) [17]. The channel opens at an acidic pH to transport urea and closes at a neutral pH to prevent alkalization [22].

Although labeling of detergent-purified wild-type KvAP has been previously reported [23], we faced significant challenges in replicating the procedure. To address this challenge, we propose a robust alternative using native nanodiscs, which encapsulate membrane proteins within a nanoscale lipid bilayer stabilized by a belt of membrane scaffold protein (MSP) [24], amphiphilic copolymers [25,26] or small-molecule amphiphiles [27]. Native nanodiscs maintain the native-like lipid environment of the protein, preserving its conformation [28,29]. Thus, they are likely to ensure the accessibility of surface-exposed cysteine residues for labeling [30]. To demonstrate this approach, we used Glyco-DIBMA, a non-aromatic copolymer with improved properties for extracting and stabilizing membrane proteins within a native-like lipid-bilayer environment [31,32]. This polymer-assisted purification method is entirely detergent-free, distinguishing it from membrane scaffold-protein-based techniques, which require detergent for the initial solubilization of membrane proteins [33]. Glyco-DIBMA overcomes the limitations of aromatic SMA and outperforms its parent polymer DIBMA in terms of lipid solubilization and protein extraction. Importantly, Glyco-DIBMA forms nanodiscs with rather narrow size distributions, ensuring consistency and reliability for applications that demand uniform nanodisc populations, such as structural studies or high-resolution imaging techniques [31]. Moreover, Glyco-DIBMA’s glycosylated hydrophilic groups and non-aromatic hydrophobic groups render it chemically inert [31]. A large variety of experiments require the use of fluorescent polymer nanodiscs. Here, only a few examples can be provided: fluorescent nanodiscs can be used to probe protein–lipid interactions [34], the binding of small molecules [35], and the oligomeric organization of membrane proteins [36].

In this study, we demonstrate the site-specific fluorescent labeling of nanodisc-embedded KvAP and *Hp*UreI using thiol–maleimide chemistry. Labeling was targeted to native cysteine residues (C260 in KvAP) or introduced point mutations (K102C or E275C in KvAP, and L134C in *Hp*UreI). To reduce non-specific interactions with membranes, we used Alexa Fluor 647 (AF647), a relatively hydrophilic dye [37]. The labeling and purification efficiencies of the proteins were evaluated by SEC and fluorescence correlation spectroscopy (FCS) [38,39], establishing nanodisc-assisted fluorescent labeling as a reliable method for membrane protein studies.

## 2. Materials and Methods

*Detergent-assisted purification and labeling of HpUreI*. An overnight culture of *Escherichia coli* (*E. coli*) C43 harboring the plasmid encoding the L134C single-cysteine mutant of *Hp*UreI was diluted 1:20 into 6 L of 2x Yeast Extract Tryptone (2xYT) medium supplemented with 100 µg/mL ampicillin. Cells were grown at 30 °C with shaking at 180 rpm until the optical density at 600 nm (OD 600) reached 0.6–0.8, at which point protein expression was induced by adding 1 mM isopropyl-β-thiogalactoside (IPTG) and continuing incubation overnight.

Cells were harvested by centrifugation at 12,000× *g* for 10 min, and the pellet was resuspended in 40 mL of lysis buffer (150 mM NaCl, 50 mM Na_2_HPO_4_, pH 7.4) containing cOmplete Mini EDTA-free protease inhibitor (Merck, Darmstadt, Germany). Lysis was performed with an EmulsiFlex-C5 (Avestin, Ottawa, Canada) at 15,000 psi for three cycles. Unlysed cells and debris were removed by centrifugation at 50,000× *g* for 30 min, followed by a second centrifugation at 100,000× *g* for 60 min to pellet the membrane fraction.

The membrane pellet was resuspended in 80 mL of resuspension buffer (150 mM NaCl, 50 mM Na_2_HPO_4_, pH 7.4), and either n-dodecyl-β-D-maltoside (DDM) or N,N-dimethyl-n-dodecylamine N-oxide (LDAO) was added to a final detergent concentration of 1% to solubilize the membrane proteins. After 60 min of shaking at 4 °C, insoluble components were removed by centrifugation at 100,000× *g* for 30 min. The supernatant was then incubated with cobalt beads equilibrated with a buffer (150 mM NaCl, 50 mM Na_2_HPO_4_, 0.03% DDM or 0.6% LDAO, pH 7.4) for 2.5 h at 4 °C with shaking.

The slurry was transferred to a gravity column and sequentially washed with a wash buffer (150 mM NaCl, 50 mM Na_2_HPO_4_, 0.03% DDM or 0.6% LDAO, 10 mM imidazole, pH 7.4), followed by 150 mL of a high-salt wash buffer (750 mM NaCl, 50 mM Na_2_HPO_4_, 0.03% DDM or 0.6% LDAO, pH 7.4) and 100 mL of an equilibration buffer.

Three labeling methods were tested:Method: after the final wash step, 10 µL of 10 mM AF647 maleimide dissolved in dimethyl sulfoxide (DMSO) was added to the sample with or without tris(2-carboxyethyl)phosphine (TCEP) and incubated overnight at 4 °C on a roller shaker.Method: following the same initial steps as Method 1, the sample was instead incubated with AF maleimide at room temperature for 1 h.Method: An on-column labeling method using vigorous TCEP treatment was tested. To minimize dilution, 200–500 µM TCEP was added directly to the wash buffer. The sample was incubated twice on the column with TCEP for 15 min at room temperature, using the first and last 5 mL of the wash buffer. This step was followed by a high-salt wash, after which 10 µL of 10 mM AF647 maleimide in DMSO was added. AF647 maleimide is widely used for specific protein labeling. For instance, its lack of nonspecific labeling has been demonstrated in studies determining membrane protein stoichiometry through stepwise photobleaching of dye-labeled subunits [40]. This observation aligns with a systematic investigation of fluorophore interactions with lipid membranes [37], which identified Alexa 647 as one of the least hydrophobic fluorophores, exhibiting a very low interaction factor. This property significantly reduces nonspecific protein labeling.

The labeled protein was eluted from the beads in five 1 mL rounds of the elution buffer (150 mM NaCl, 50 mM Na_2_HPO_4_, 0.03% DDM or 0.6% LDAO, 250 mM imidazole, pH 7.4), with each round incubating for 3–5 min. The 6xHis tag was removed by overnight treatment with 8 units/mL of Pierce HRV 3C protease. The sample was concentrated to 1.5–2 mL and eluted with an equilibration buffer.

SEC was performed on an Äkta Pure System for fast protein liquid chromatography with a Superdex 200 increase 10/300 column from Cytiva (Merck) equipped with a UV/fluorescence detector to evaluate protein quality, quantity, and labeling efficiency. Previously purified proteins with known molar masses served as controls to estimate protein mass and oligomeric state. Standard runs were taken in different buffers and at different time points. The standard used was the BioRad Gel Filtration Standard from Bio-Rad Laboratories (Vienna, Austria). The run data were used to generate a standard curve, which in turn was used to estimate the size of the molecules or complexes corresponding to the peaks.

*Detergent-assisted purification and labeling of KvAP*. KvAP wild-type and single-cysteine mutant variants (K102C or E275C) were overexpressed in *E. coli* C43 cells using a pET21a-derived expression vector. Cultures were grown for 4 h in 2 L of 2x Yeast Extract Tryptone (2xYT) medium, followed by induction with 1 mM IPTG. Collected cell pellets were resuspended in extraction buffer (100 mM KCl, 25 mM Tris, 1 mM MgCl_2_, pH 8.0) supplemented with protease inhibitors. Cells were lysed using a French pressure cell press, using the same pressure settings as for *Hp*UreI.

Following lysis, cell debris was removed by centrifugation at 6500× *g* for 10 min at 4 °C, and the membrane fraction was pelleted by centrifugation at 100,000× *g* for 2 h at 4 °C. The membrane pellet was resuspended in an extraction buffer containing 4% (*w*/*v*) N-decyl-β-maltoside (DM) and 2 mM TCEP, then solubilized by gentle shaking for 2 h at 4 °C. Insoluble material was removed by centrifugation at 100,000× *g* for 2 h, and the supernatant was subjected to affinity chromatography using Ni-NTA agarose.

On-column labeling was conducted as described in Method 3 for *Hp*UreI, with AF647 maleimide. Purified protein complexes were eluted from the Ni-NTA beads using an SEC buffer (100 mM KCl, 25 mM Tris, pH 8) containing 0.25% (*w*/*v*) DM and 400 mM imidazole. Elution fractions were pooled, concentrated to 500 µL, and further purified by SEC to assess purity, complex stability, and labeling efficiency.

*Reconstitution of Detergent-Purified KvAP into Lipid Vesicles*. KvAP was reconstituted into *E. coli* polar lipid extract (Avanti Polar Lipids, Alabaster, USA) vesicles prepared with 0.5% (*w*/*v*) DM, using a protein/lipid mass ratio of 1:200. PLE was selected for its broad accessibility and its ability to closely mimic the general lipid composition of bacterial membranes. This makes it an ideal choice for reconstitution studies aimed at capturing native-like environments for membrane proteins. Excess detergent was removed by incubating the sample overnight at 4 °C with Bio-Beads (SM2, Bio-Rad) under gentle mixing. The resulting turbid suspension was pelleted by centrifugation at 61,177× *g* for 1.5 h at 4 °C. The pellet was resuspended in a buffer (10 mM HEPES, 15 mM KCl, 10% (*v*/*v*) glycerol, pH 7.5) and extruded through a 100 nm filter. The reconstituted sample was then aliquoted, flash-frozen in liquid nitrogen, and stored at –80 °C.

*Nanodisc-Assisted Purification and Labeling of HpUreI and KvAP*. Protein overexpression and cell harvest for both *Hp*UreI and KvAP were conducted as described above. The cell pellets were lysed in 40 mL of a standard buffer (150 mM NaCl for *Hp*UreI or 150 mM KCl for KvAP, 50 mM Tris, pH 7.4) containing protease inhibitors, 2.5 mM MgSO_4_, and 5 mg/mL DNase. Lysis was performed with a French pressure cell press using the same settings as above, followed by the same centrifugation steps used for detergent-assisted purification. The membrane fraction was resuspended in the standard buffer at a concentration of 50–150 mg/mL and mixed with Glyco-DIBMA in a 1:1 mass ratio, with both components dissolved in the same buffer (pH adjusted to 7.4–8). To enhance solubilization, 4 mM MgCl_2_ was added, and the sample was incubated overnight at 18–20 °C with shaking. The next day, insoluble material was removed by centrifugation at 50,000× *g* for 30 min. The supernatant containing solubilized membrane proteins was incubated overnight at 4 °C with 1 mL of Ni-NTA beads equilibrated in the standard buffer.

The slurry was transferred to a gravity column, allowed to settle for 5 min, and washed sequentially with 100 mL of a TCEP-containing standard buffer, followed by a wash with a TCEP-free buffer. For labeling, 10 µL of 10 mM AF647 maleimide dissolved in DMSO was added to 1 mL of the buffer atop the beads, and the mixture was incubated either for 1 h at room temperature or overnight at 4 °C. Unbound dye was removed by washing with 50 mL of the standard buffer. The labeled membrane proteins were eluted in five 400 µL rounds of the elution buffer (standard buffer with 0.8 M imidazole) and further purified using SEC.

*Electrophysiology.* Solvent-depleted planar lipid bilayer (PLB) experiments were conducted following established procedures [41,42]. After PLB formation, vesicles containing reconstituted KvAP were added near the PLB, and fusion was facilitated by a salt concentration gradient across the membrane. Vesicles were introduced to the ‘intracellular’ side with 15 mM KCl, while the ‘extracellular’ side contained 150 mM KCl, both buffered with 10 mM HEPES at pH 7.4. Voltage-clamp measurements were performed in whole-cell mode using an EPC 9 patch-clamp amplifier (HEKA Elektronik, Reutlingen, Germany). Currents were acquired at 25 kHz and filtered at 10 kHz using Bessel filters. The extracellular side of the channel was set to ground, in accordance with electrophysiological conventions. After channel appearance, both sides of the measurement chamber were adjusted to 150 mM KCl.

*Fluorescence correlation spectroscopy*. FCS was used to confirm successful protein labeling and estimate the number of fluorescent labels per oligomer. Measurements were performed with a laser scanning microscope equipped with an FCS extension (MicroTime 200 PicoQuant, Berlin, Germany). As AF647 maleimide-labeled proteins diffused through the diffraction-limited observation volume, temporal fluorescence intensity fluctuations were detected by avalanche photodiodes after passing through a band-pass emission filter [43]. We obtained the number (*N*) of diffusing particles from the autocorrelation function *G*(*τ*) of the fluorescence temporal signal of labeled proteins (Equation (1)):(1)$$G\left(\tau \right)=\frac{1}{N}{\left(1+\frac{\tau }{\tau_{D}}\right)}^{-1}*{\left(1+\frac{\tau }{\kappa^{2}\tau_{D}}\right)}^{-0.5}$$
where *N* is the average number of fluorescent particles in the volume, $\tau_{D}$ is the diffusion time, and *κ* denotes the axial-to-radial aspect ratio of the confocal volume. Knowledge of the size of the focal volume allows determination of the particle concentration and diffusion coefficient from *N* and $\tau_{D}$, respectively.

## 3. Results and Discussion

To evaluate the efficacy of our nanodisc-assisted labeling approach in preserving native protein structure while enabling site-specific fluorescent labeling, we conducted a series of experiments comparing traditional detergent-based methods with nanodisc extraction for two challenging membrane proteins, KvAP and *Hp*UreI.

### 3.1. Detergent-Assisted Purification of KvAP and Labeling Challenges

Wild-type KvAP was initially overexpressed in *E. coli*, purified into detergent micelles, and subjected to on-column labeling with AF647 maleimide dye. SEC confirmed the expected homotetrameric conformation with a UV detection peak at an elution volume of 10–12.5 mL (Figure 2A). However, SDS-PAGE analysis of the peak fraction of SEC revealed that fluorescent labeling was unsuccessful, as no fluorescence signal was detected (Figure 2B). To confirm that the lack of successful labeling was not due to protein aggregation or misfolding, KvAP was reconstituted into unilamellar vesicles made from *E. coli* polar lipids, which were osmotically fused with planar lipid bilayers (PLBs), as previously described [41]. Single-channel electrophysiology confirmed the functionality of purified KvAP (Figure 2C), with voltage ramp experiments demonstrating K^+^ selectivity (Figure 2D). Hence, our experiment demonstrated that the protein retained its functionality throughout the purification and reconstitution process. Thus, the primary factor preventing effective labeling was the inaccessibility of the cysteine residues in detergent micelles.

To investigate whether only the native cysteine of KvAP is inaccessible to maleimide dye in micelles, it was mutated to alanine, and cysteine residues were introduced at position K102 or E275 (Figure 1A). Structural predictions indicate that these residues are located at the protein surface and should be accessible for maleimide dye binding. SEC confirmed the successful purification of tetrameric KvAP for both mutants. However, labeling attempts with maleimide dye were unsuccessful for both K102 and E275 (Appendix A).

An alternative approach could involve reconstituting membrane proteins from detergent micelles into liposomes, followed by performing fluorescent labeling. The use of liposomes for labeling membrane proteins presents several challenges. First, the orientation of reconstituted proteins in liposomes often results in a 50:50 distribution, meaning that a significant fraction of the labeling tags may face the interior of the liposomes, rendering them inaccessible to membrane-impermeable dye molecules. Second, removing unbound dye requires stringent washing steps, which often necessitate specialized equipment and careful experimental design when working with liposomes. These additional steps can also risk compromising the integrity of the liposomes, further complicating the process. In contrast, our approach using polymeric nanodiscs avoids these limitations, offering an efficient and accessible alternative for precise protein labeling in native-like environments.

### 3.2. Nanodisc-Assisted Purification of KvAP and Successful Labeling

To ensure the accessibility of surface-exposed cysteine residues, both wild-type and point mutants of KvAP were purified from *E. coli* cell membranes into native nanodiscs using the copolymer Glyco-DIBMA. The basic purification protocol was similar to the detergent-assisted method, with on-column labeling performed during affinity chromatography. Following elution from the Ni-NTA beads, the proteins were further purified by SEC in the standard buffer.

A primary UV absorption peak appeared between 10 mL and 12.5 mL of elution volume, consistent with the results obtained from detergent-assisted purification. The similar elution volumes observed for the protein in nanodiscs and detergent micelles suggest that their sizes are comparable. Nanodiscs formed with polymers such as Glyco-DIBMA are relatively compact, with hydrodynamic radii closely matching those of protein-detergent micelle complexes. In contrast to the detergent method, however, the fluorescence peaks now coincided with the UV signal, indicating high labeling efficiency for the wild-type protein (Figure 3).

Successful labeling of the KvAP mutants was also confirmed (Appendix A). To further analyze the oligomeric state of the nanodisc-embedded proteins, the eluted fractions were subjected to FCS.

FCS measurements of SEC-purified KvAP confirmed both successful labeling and the preservation of their native oligomeric states (Figure 4A and Figure 5). Control measurements of free AF647 maleimide dye established a baseline fluorescence intensity of 9 kHz. Wild-type KvAP and its cysteine mutants (K102C and E275C) were measured using the same laser power, showing an increase in molecular brightness that correlated with the oligomeric states, assuming the labeling of all monomers within each oligomer. Specifically, molecular brightness increased to 46 kHz and 42 kHz for wild-type KvAP and its K102C and E275C mutants, respectively. These values match the expected oligomeric forms, reflecting an approximately 4-fold increase for tetrameric KvAP (Figure 4B). FCS autocorrelation curves were fitted with a single-component equation, confirming that the fluorescence signal originated from a single fraction of fluorescently labeled oligomeric membrane protein. The purified nanodiscs were estimated to have a diameter of approximately 20 nm across all three KvAP constructs, providing sufficient space to accommodate the tetrameric protein.

### 3.3. Nanodisc-Purified HpUreI Can Be Fluorescently Labeled While Preserving Its Oligomeric State

Because *Hp*UreI lacks native cysteine residues, a leucine at position 134 was mutated to cysteine to enable site-specific labeling. *Hp*UreI was initially purified in DDM and labeled using both in-solution methods (labeling methods 1 and 2) and on-column labeling (labeling method 3), with and without the reducing agent TCEP (Appendix A). SEC revealed a mixture of oligomeric forms and some aggregates; however, none of these were successfully labeled with Alexa dye. Although the addition of TCEP slightly improved cysteine accessibility, as evidenced by SEC, only aggregated and non-hexameric oligomers were labeled, while the physiologically relevant *Hp*UreI homohexamer remained unlabeled (Appendix A).

SDS-PAGE analysis indicated that switching from DDM to LDAO during purification helped preserve the native hexameric state of *Hp*UreI (Appendix A). Since TCEP is essential for cysteine labeling and the on-column method is more efficient by reducing time, buffer usage, and sample dilution, we opted for this approach. Nevertheless, despite successful purification, SEC confirmed that labeling remained unsuccessful, even with a reducing agent (Figure 6A).

To explore an alternative, *Hp*UreI L134C was also purified into native nanodiscs by extracting the protein from the cell membrane with the amphiphilic copolymer Glyco-DIBMA, with on-column labeling performed during affinity chromatography. After elution from the Ni-NTA beads, SEC analysis in standard buffer showed a primary UV absorption peak at 10.5 mL, matching the elution profile observed in detergent-based purification (Figure 4B). As with KvAP, the fluorescence peaks corresponded closely with the UV signal, indicating successful labeling.

As for KvAP, FCS measurements of SEC-purified *Hp*UreI confirmed both successful labeling and the preservation of their native oligomeric states (Figure 4A and Figure 5). *Hp*UreI L134C was measured using the same laser power as the control and the KvAP fractions, showing an increase in molecular brightness that correlated with the oligomeric states, assuming the labeling of all monomers within each oligomer. Specifically, molecular brightness increased to 60 kHz for *Hp*UreI. This value matches the expected oligomeric forms, reflecting an approximately six-fold increase for hexameric *Hp*UreI (Figure 4B). The nanodiscs were estimated to have a diameter of approximately 17 nm, offering ample space to accommodate the hexameric form of the protein. FCS autocorrelation curves were fitted with a single-component equation, confirming that the fluorescence signal originated from a single fraction of fluorescently labeled oligomeric membrane proteins.

A comparison of the FCS measurements of detergent-solubilized and nanodisc-embedded *Hp*UreI at identical laser power revealed a notable difference in labeling efficiency (Figure 5B). Unlike nanodisc-embedded *Hp*UreI, the detergent-solubilized protein exhibited no detectable fluorescent signal, resulting in an autocorrelation curve that could not be fitted.

In contrast to most polymeric nanodiscs, which exhibit broad size distributions, our Glyco-DIBMA preparations are more homogeneous. This conclusion is supported by fluorescence correlation spectroscopy (Figure 5). If the nanodiscs exhibited a broad size distribution, they would diffuse at various rates: smaller particles would diffuse faster, resulting in shorter residence times in the focal volume, whereas larger particles would diffuse slower, leading to longer residence times. Consequently, a broad size distribution would result in a composite autocorrelation function that could not be accurately described by a single diffusion time. Instead, our data fit well to a single residence time, corresponding to a single diffusion coefficient. This observation aligns with our previous findings, which demonstrated that Glyco-DIBMA nanodiscs exhibit a much narrower size distribution compared to other polymer-encapsulated nanodiscs [31].

In summary, these results clearly demonstrate that the nanodisc-assisted approach enables more robust and efficient labeling of membrane proteins compared to detergent-based methods (Figure 7). The accessibility of surface-exposed cysteine residues in polymer-encapsulated nanodiscs facilitated effective fluorophore conjugation to bilayer-embedded membrane proteins. FCS detected approximately one dye molecule per protein monomer, further validating the expected oligomeric states of both KvAP and *Hp*UreI.

## 4. Conclusions

Fluorescent labeling of membrane proteins is essential for understanding their functional roles, interactions, and structural dynamics. Organic fluorophores, prized for their favorable photophysical properties and photostability, are widely used for these applications. However, a persistent challenge is the inaccessibility of surface-exposed cysteine residues, often occluded within detergent micelles during protein purification. To overcome this limitation, we used polymer-encapsulated nanodiscs to maintain the native-like environment of membrane proteins, also preserving the accessibility of key cysteine residues for labeling.

Compared to the alternative approach of labeling the proteins after reconstitution into liposomes, protein labeling in nanodiscs offers several advantages. First, cysteine residues are accessible regardless of the protein’s orientation. In contrast, when the protein is reconstituted prior to labeling, only half of the randomly oriented proteins can be labeled, as intraluminal cysteines remain inaccessible. Second, on-column labeling of the protein in nanodiscs eliminates the need for an additional washing step, as the washing required for protein release during affinity chromatography suffices. In contrast, labeling after reconstitution necessitates an extra washing step. Furthermore, the unspecific binding of the free dye to the vesicle membrane poses a significant challenge during labeling after reconstitution. This issue arises because the protein-to-lipid ratio decreases by several orders of magnitude during reconstitution.

While genetically encoded fluorescent proteins, such as green fluorescent protein (GFP) [44] have undergone extensive development, achieving greater stability, quantum yield, and wavelength range [45], their large size can interfere with the physiological functions, trafficking, or structural dynamics of target proteins. Moreover, the tendency of some fluorescent protein tags to cause oligomerization can complicate experimental outcomes [46,47]. By utilizing cysteine-specific labeling, our approach achieves a level of labeling efficiency approaching that of genetically encoded dyes but avoids these critical drawbacks, offering a more precise tool for studying membrane protein function in a native-like bilayer environment.

Our approach simplifies the labeling process by integrating protein purification and dye conjugation into a single step. His-tagged proteins are purified within native nanodiscs using affinity chromatography, allowing the addition of fluorescent dyes during purification. This method enables the successful labeling of two membrane proteins—KvAP and *Hp*UreI—where detergent-based approaches face limitations. Crucially, the nanodisc-based strategy facilitates efficient fluorescent labeling while preserving the proteins’ native oligomeric states.

This robust, versatile method marks a significant advancement in membrane protein labeling, ensuring structural integrity under near-native conditions. It opens new avenues for studying a diverse range of membrane proteins, providing a powerful tool for investigating complex biological systems at the molecular level.

## Figures and Tables

**Figure 1 biomolecules-15-00254-f001:**
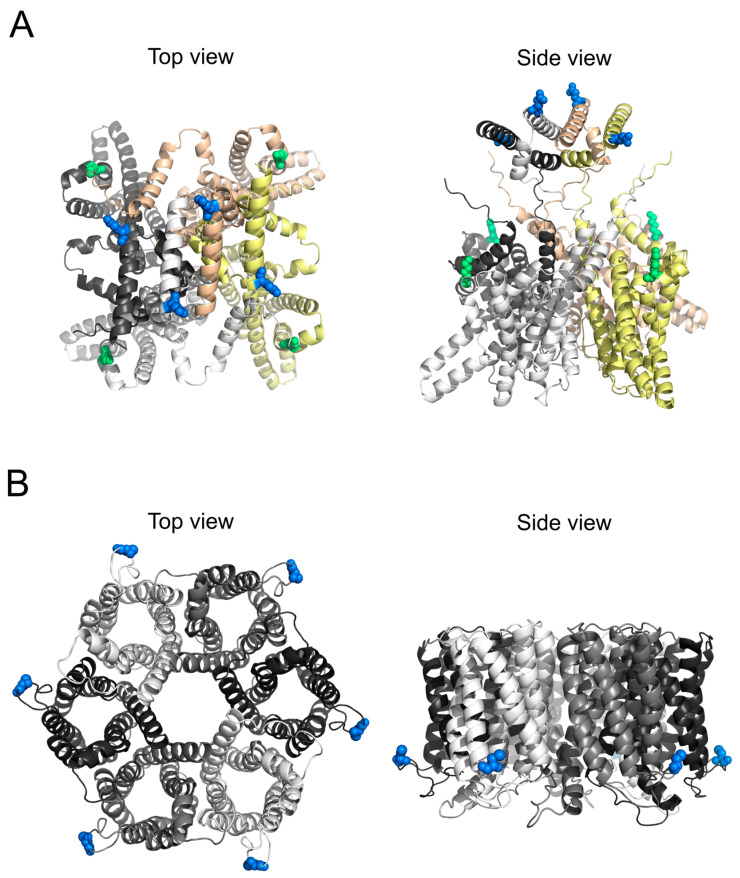
Structural representation of the trimeric voltage-gated sodium channel KvAP (**A**) and the hexameric proton-gated urea channel *Hp*UreI (**B**). Cysteine residues, critical for fluorescent labeling, are highlighted to emphasize their spatial distribution. Both proteins have been modeled with AlphaFold to visualize elements missing in the experimentally derived pdb structures (KvAP: 6UWM, *Hp*UreI: 6NSK).

**Figure 2 biomolecules-15-00254-f002:**
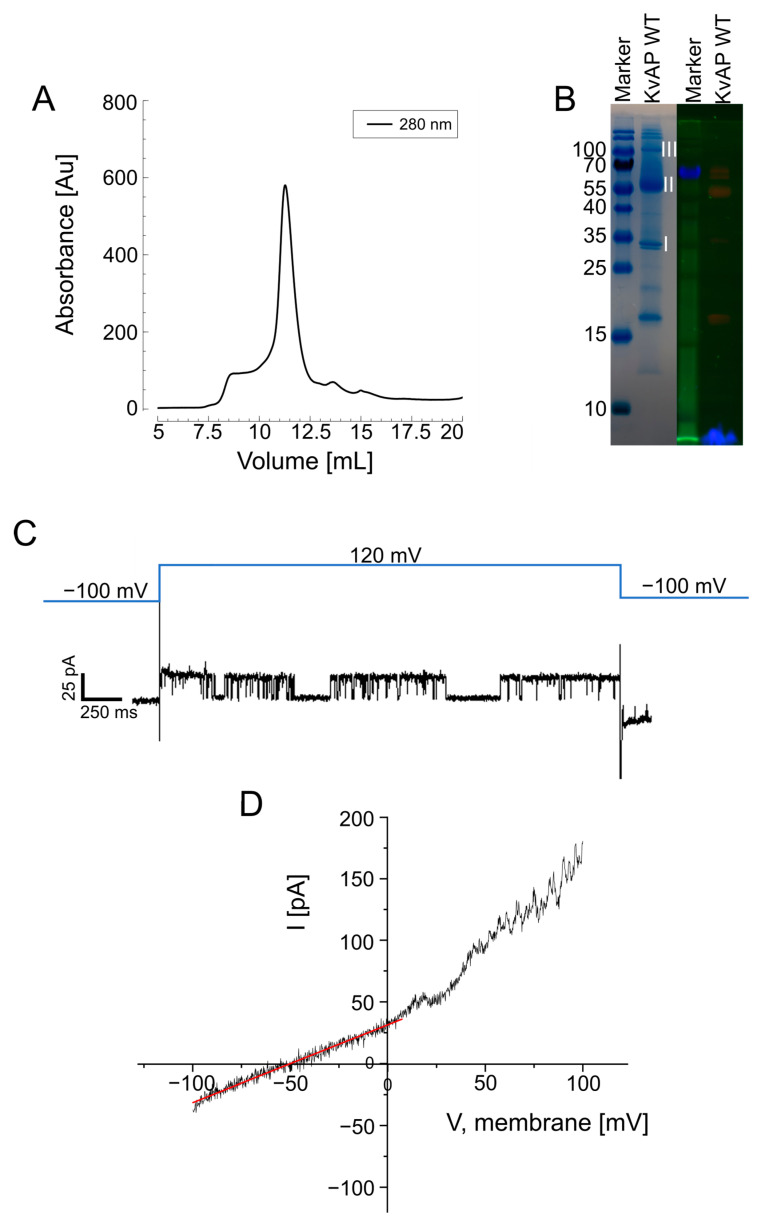
(**A**) SEC of KvAP solubilized in 0.25% (*w*/*v*) DM micelles. The elution peak between 10 mL and 12.5 mL corresponds to the homotetrameric fraction of KvAP. (**B**) SDS-PAGE analysis of the KvAP/micelle mixture. The gel was imaged at a BioRad GelDoc-System. Left panel: Coomassie-stained gel showing protein bands. Right panel: fluorescent gel image of AF647 maleimide labeling. I: monomer, II: dimer, III: tetramer. For excitation at 647 nm, an LED was used, and the emission filter was set at (695 ± 27) nm. (**C**) Single-channel recording of KvAP reconstituted into solvent-depleted planar lipid bilayers (PLBs) formed from *E. coli* polar lipid extract (PLE) in 150 mM KCl and 10 mM HEPES, pH 7.4. (**D**) K^+^ selectivity test. The voltage was ramped from −100 mV to +100 mV over 100 ms in whole-cell configuration. The K^+^ gradient (15 mM KCl outside, 150 mM KCl inside) resulted in a reversal potential of −50 mV, as determined from a linear fit to the data (red line), i.e., a value close to the expected Nernst potential of K^+^.

**Figure 3 biomolecules-15-00254-f003:**
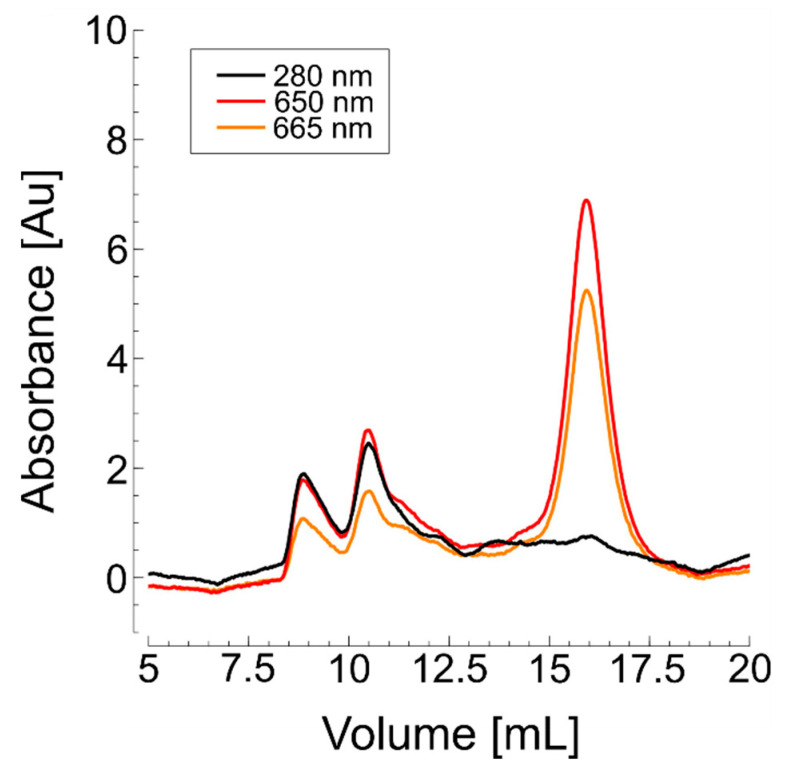
SEC of nanodisc-purified wild-type KvAP labeled with AF647 maleimide. The black curve represents UV absorbance, while the orange and yellow curves indicate fluorescence emission at 650 nm and 665 nm, respectively. Tetrameric KvAP elutes between 10 mL and 12.5 mL.

**Figure 4 biomolecules-15-00254-f004:**
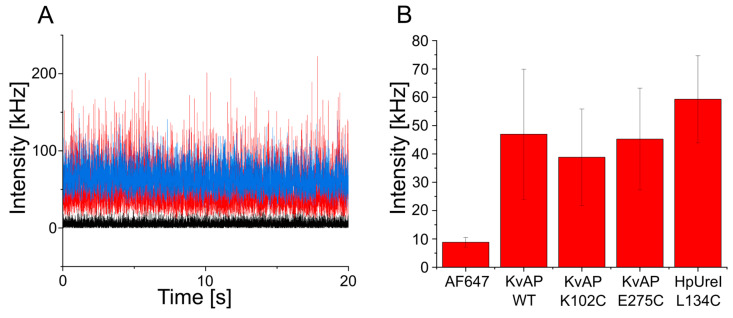
(**A**) The fluorescence intensity time traces show Alexa Fluor 647-maleimide (black), wild-type KvAP (red), and *Hp*UreI L134C (blue) measured under the same conditions, illustrating the differences in fluorescent signals between the free dye and the labeled proteins. (**B**) Quantitative comparison of molecular brightness for different fractions of purified membrane proteins (wild-type KvAP, KvAP K102C, KvAP E275C, and *Hp*UreI L134C) relative to the free Alexa Fluor 647-maleimide dye, which serves as a reference standard. The molecular brightness values reflect the labeling efficiency and oligomeric states of the analyzed proteins.

**Figure 5 biomolecules-15-00254-f005:**
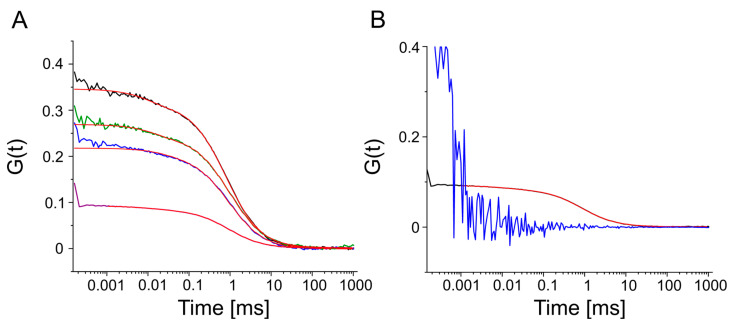
FCS of labeled KvAP and *Hp*UreI. (**A**) Autocorrelation functions of nanodisc-embedded tetrameric KvAP wild-type (black, $\tau_{D}\;$ = 877 µs, concentration = 3.23 nM), mutant K102C (blue, $\tau_{D}$ = 887 µs, concentration = 3.72 nM), mutant E275C (green, $\tau_{D}$ = 846.5 µs, concentration = 2.96 nM), and hexameric *Hp*UreI L134C (purple, $\tau_{D}$ = 853.5 µs, concentration = 8.24 nM) after SEC. The red curves represent fits of Equation (1) to the experimental data. The listed $\tau_{D}$ and concentration values are fit parameters, where the actual fit parameter N has been converted into concentration based on the known size of the focal volume. (**B**) Labeling efficiency of hexameric *Hp*UreI purified in detergent (blue) or Glyco-DIBMA (black). The autocorrelation functions confirm successful labeling of nanodisc-embedded *Hp*UreI, whereas detergent-solubilized *Hp*UreI remained unlabeled.

**Figure 6 biomolecules-15-00254-f006:**
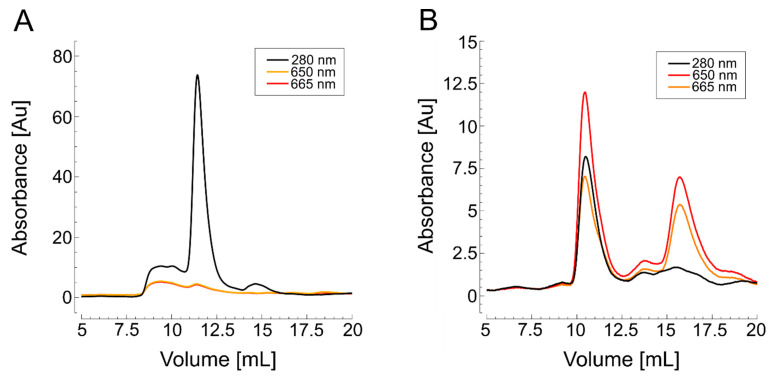
SEC of purified *Hp*UreI labeled with AF647 maleimide. The black curve represents UV absorbance, while the orange and yellow curves represent fluorescence emission at 650 nm and 665 nm, respectively. The native homohexameric fraction is indicated by the UV peak at an elution of 12–12.5 mL. (**A**) Detergent-assisted purification with LDAO. No fluorescent signal was detected, indicating unsuccessful labeling. (**B**) Nanodisc-assisted purification of *Hp*UreI using Glyco-DIBMA. Fluorescent peaks coincide with the UV peak, indicating successful labeling of hexameric *Hp*UreI L134C.

**Figure 7 biomolecules-15-00254-f007:**
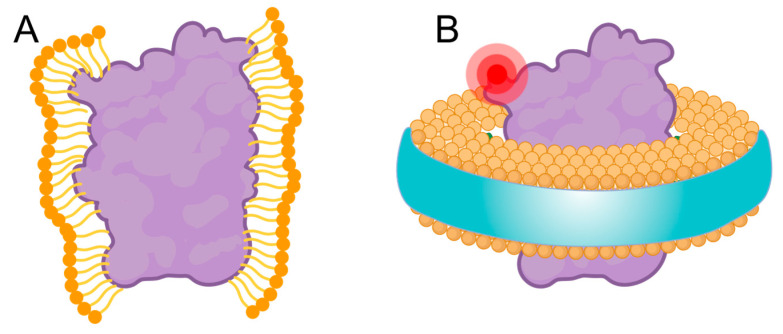
Protein labeling scheme in (**A**) detergent micelles and (**B**) nanodiscs. (**A**) Dissolution of membrane proteins in detergent micelles can obscure the labeling site, reducing the efficiency of covalent attachment via chemical reactions. (**B**) Extraction of proteins using native nanodiscs preserves the accessibility of the labeling site, thereby increasing the efficiency and robustness of fluorescent labeling.

## Data Availability

The research data are available on Zenodo (DOI: 10.5281/zenodo.14620507).

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
