# Peer review of "Enhanced Site-Specific Fluorescent Labeling of Membrane Proteins Using Native Nanodiscs"

_biomolecules, 2025, doi:10.3390/biom15020254_

Round 1
Reviewer 1 Report
Comments and Suggestions for Authors
Overall, this is a really nice paper that demonstrates an approach that will be useful more widely in the field. I have highlighted a number of small points below and also a question which the authors may be able to address but may also prompt some additional controls to be included.
Typos:
· Line 130 should be “h at” not “hat”.
Materials and methods:
· State the concentration of ampicillin and IPTG rather than the dilution.
· State centrifuge speeds at rcf/g rather than rpm as the force depends on the rotor diameter.
· State which protease inhibitors were used.
· State which SEC column was used.
Results:
· Fluorescent gel imaging (Fig. 1B) – no details are given as to the excitation and emission wavelengths used here and is it possible that this is just outside the range for the fluorophore (which has quite a high excitation wavelength compared to most)?
· Which fractions were run on the gel in 1B?
· I have a question / concern about the labelling in nanodiscs. Although the authors have rightly said that the fluorophore chose in relatively hydrophilic, it could still be integrating into the bilayer of the disc. I couldn’t see a control of e.g. HpUre1 without a cysteine in it, in a nanodisc, plus fluorophore? The absence of labelling from this would give more confidence in the data. Alternatively, the nanodisc-encapsulated protein could be isolated and mass spec used to confirm. Even easier, the authors could run an SDS-PAGE as in Fig.1 and check if the protein band is fluorescent? I think one of these options (or another you think of that shows the same thing) is essential to demonstrate that the label isn’t just going into the disc. Further to this, I appreciate that the FCS was used to look at the oligomeric state, but how can the authors be sure that none of the fluorescence observed is due to the dye being in the disc and not attached to the protein. FCS is not my field so this may be possible to explain! Essentially, can you be sure there is no labelling of the lipid. This would be important for others to take this approach forwards.
· The last paragraph of the results seems like it belongs in the intro or conclusion to me.
Reviewer 2 Report
Comments and Suggestions for Authors
The paper addresses the problem of labeling membrane proteins at cysteine residues, which may be difficult in detergent micelles. The authors use two membrane proteins as an example and they incorporate those membrane proteins in polymer nanodiscs.
1. The paper is quite messy and disorganized. The introduction should at least introduce basic origin and functions of the proteins tested, and describe experiments that require the use of fluorescent polymer nanodics. Lastly, additional data that supports oligomeric size for this proteins should be provided.
2. While I have no doubt that the authors have been able to produce those nanodiscs, in my experience, polymeric nanodiscs produce heterogeneous preparations in size, whereas preparations obtained with MSP are more homogeneous preparations, eg, for Cryo-EM. The paper does not appear to show any control for the size of these nanodiscs, or any comment regarding their application.
3. The authors also should explain with specific examples why this method is useful, since I can only see general comments. Depending on the application, membrane proteins can be labeled simply using liposomes. Of course the problem is unfavourable asymmetric or symmetric insertion, but this point should be elaborated in the discussion.
4. Why is the lipid chosen E coli polar lipid extract? Are these proteins from E Coli? where other lipid types tested?
5. In Fig. 1B, labels for oligomers in the gel are confusing. ‘I’ should be monomer, ‘II’ dimer, and so on. There are two lanes in the right panel, but only one lane is described.
6. Lane 193. Since the protein cannot be labeled in micelles, ‘to address this issue’ …the protein is placed in planar bilayers to measure its function (?). This does not make sense.
7. Lane 226. The eluted fractions were subjected to FCS... but the next section 3.3 has nothing to do with FCS.
8. Lane 247. Refers to SEC and FCS for Fig. 3A, but Fig 3 has no FCS.
9. Figure 2. Why is that when the protein is incorporated into nanodiscs, the elution volume seems to be the same as when the protein is incorporated in detergent micelles? What is the size of these nanodiscs compared to the size of the micelles? How much lipid do they contain?
10. DLS to determine how homogeneous is the nanodisc preparation is also not shown. Depending on the application, this is important.
11. In part 3.4, fluorescence intensity (how is intensity measured in Hz?) is used to assess oligomeric size, but data for these intensities is not shown. Intensity may be affected by environment. Other supporting techniques should be used.
12. FCS was used to confirm oligomeric size, based on the number of fluorescent labels per oligomer. The authors say that a single component was used to fit the data, but the fit parameters, or the fitted curve are not shown. Diffusion time constant may be used to determine size of the nanodiscs.
13. In figure 4B, the protein HPUreI is used when detergent solubilized. If this protein could not be labeled in detergent, how is a non fluorescent protein measurable in FCS?
14. I could not access the Suppl file.
Reviewer 3 Report
Comments and Suggestions for Authors
Line 101, is it a typo that 10 ‘mL’ of 10 mM AF647? If this was correct, please provide the total volume of the sample for adding this much of AF647 dyes and the final concentration of DMSO in the reaction solution.
In Figure 1, panel A shows a nice single peak of tetrameric proteins. However, panel B shows several different oligomeric species - tetramers, dimers, and monomers – and the dimers are the major portion of the purified proteins. Is it because the sample was not fully reduced before running the SDS-PAGE? How could the authors confirm that the peak in panel A is tetrameric conformation?
Related to the question above, the authors did not address what SEC column they used for purification. Please provide this information so that the molecular size of the peak in Figure 1A can be estimated.
Line 213, there is no material for Figure S1 in the manuscript package.
Line 225, there is no material for Figure S2 in the manuscript package.
In Figure 2, the trace contains several different oligomeric species and there is no biochemistry data to confirm the result. The authors should show an SDS-PAGE result for the nanodisc reconstituted protein sample.
Lines 237, 242, and 248, there is no material for Figure S3 in the manuscript package.
Line 272-275, I do not see evidence supporting this statement. The authors should present proper results for this.
The authors chose several residues that were said to be surface-exposed in the target proteins. However, there is no information about which structures they used for the experimental designing or how they assumed these residues are surface-exposed. To understand the rationale of this study, I suggest that the authors have to include a figure with structures showing the exact location of those residues. If the authors used PDB coordinates, please provide the codes, or if they used AlphaFold models, please describe them accordingly.
The authors presented the altered labeling of the surface-exposed cysteine residues only with the example of glyco-DIBMA. It is unclear how these two unrelated proteins could bring the same results. Can the authors test other nanodisc to demonstrate this is not the effect of glyco-DIBMA but more general characteristics with a native-like lipid condition?
Reviewer 4 Report
Comments and Suggestions for Authors
In this manuscript, the authors determined to use organic fluorophores to label two challenging-to-label membrane proteins, KvAP and HpUreI. Recombinant membrane proteins purified with detergent tend to bury the cysteine residues inside the detergent micelles, which increases the difficulties of site-specific labeling. The authors therefore purified the two membrane proteins by the nanodisc Glyco-DIBMA and successfully labeled the proteins. SEC chromatography, solvent-depleted planar lipid bilayer experiment, and fluorescence correlation spectroscopy were performed to characterize the labeling efficiency. This technique might have the potential to apply on a variety of membrane proteins for their fluorescently labeling in a close-to-native form. This manuscript is overall in a good state with logical experimental design, reasonable data interpretation, and insightful discussion. I only have a few questions that would like the authors to address further:
1. Can the authors talk more about the rationale of the point mutations (point mutations (K102C or E275C in KvAP, and L134C in HpUreI)? I am aware that the mutated cysteine residue has to be not buried in the lipid bilayer of the nanodisc. However, I am sure there should be plenty of amino residues exposed to the surface. Will those mutations affect the protein activity or conformation?
2. Lane 80, besides the dilution fold of IPTG addition, please also add the stock concentration of IPTG or the final concentration of it.
3. Lane 101, it should be “10 µl of 10 mM AF647 maleimide” instead of “10 ml of 10 mM AF647 maleimide”.
4. Lane 140, “was used” should be deleted.
5. In Figure 1, on the SDS-PAGE result, it seems the KvAP dimer band is the strongest, but on the SEC spectrum, the homotetrameric fraction of KvAP, whose elution peak is between 10 ml and 12.5 ml, is the highest peak. Could the authors explain why the two results do not match each other?
6. In Figure 2, other than the tetrameric KvAP peak between 10 ml and 12.5 ml, could the authors also identify the peak between 7.5 ml and 10 ml, and the peak between 15 ml and 17.5 ml? I am also wondering whether the nanodisc-KvAP complex have the same particle size as the detergent-KvAP micelle. If not, how can the authors be sure that the same peak between 10 ml and 12.5 ml in the detergent-assisted purification and in the nanodisc-assisted purification are both the tetrameric KvAP elute?
7. In my understanding, different nanodiscs have different capacities due to the length of the copolymer that constrains the phospholipids. Could the authors explain what is the reason of choosing the Glyco-DIBMA in this case? And may the monomer, dimer, and tetramer of the target membrane protein be captured differently by the same Glyco-DIBMA nanodisc?
Round 2
Reviewer 3 Report
Comments and Suggestions for Authors
The authors provided the information for the size-exclusion chromatography column and clarified that the size of the purified protein was correct based on the calibrated measurement. This is fundamental information and should be provided previously to properly assess their purification data. I do not have further concerns about the quality of proteins.
Despite fluorescence correlation spectroscopy can be useful for the quantitative analysis of the proteins, it does not tell the quality of the protein, i.e., whether the target protein was correctly incorporated into the nanodisc. Since the authors already assessed the purified protein in detergent condition with SDS-PAGE, they should be able to check the nanodisc incorporated protein sample in the same way. In some cases, these kinds of synthetic nanodiscs do not successfully embed the protein during the reconstruction process. Also, it is not uncommon that the detergent-assisted proteins still can survive in a non-detergent buffer condition especially if they were extracted with a strong detergent such that was used in this study. As a biochemistry aspect, the authors still do not show direct evidence of whether the resulting sample is properly incorporated into the nanodisc for their following experiment.
For the newly added Figure 5 in the revised manuscript, there is no difference between the KvAP WT and the cysteine mutants. Instead, the fluorescence intensity of K102C looks weaker than the WT. This raises a concern that the K102C and E275C mutants should be brighter than the WT because they have additional sites for fluorescence dye conjugation. Also, the authors said that they mutated the native cysteine of KvAP to alanine in lines 259-260. As such, it is unclear how the KvAP WT can be labeled in this condition or if there are still surface-exposed reactive cysteines in the WT. Please describe in the text more clearly.
Regarding the author’s response 8, the authors misunderstood the point of the question. The question was not about their fluorescence intensity, it was about whether there was a possible artifact by this particular chemical species, glyco-DIBMA. If the altered fluorescence labeling is caused by the use of nanodisc rather than detergent, this should work in other SMA or traditional lipid nanodisc. This is the main scope of this paper, but the authors only tried one chemical species to support this idea.
Author Response
We thank the reviewer for the valuable feedback. Below, we address each comment in detail.
Comment 1
The authors provided the information for the size-exclusion chromatography column and clarified that the size of the purified protein was correct based on the calibrated measurement. This is fundamental information and should be provided previously to properly assess their purification data. I do not have further concerns about the quality of proteins.
Response:
We thank the reviewer for confirming that we have addressed the concern regarding the quality of the proteins.
Comment 2
Despite fluorescence correlation spectroscopy can be useful for the quantitative analysis of the proteins, it does not tell the quality of the protein, i.e., whether the target protein was correctly incorporated into the nanodisc.
Response:
We respectfully disagree with the reviewer’s concern. Our nanodiscs do not remove the protein from its native environment. Instead, they encapsulate the protein together with its surrounding lipid belt. Thus, the nanodiscs assemble around the protein and its annular lipids without requiring incorporation. We have demonstrated this principle in our previous studies (e.g., Danielczak et al., Nanoscale, 2022), where we showed that Glyco-DIBMA efficiently extracts membrane proteins while preserving their intact lipid-bilayer environment.
Since the authors already assessed the purified protein in detergent condition with SDS-PAGE, they should be able to check the nanodisc incorporated protein sample in the same way.
Response:
We would like to clarify that extracting the protein from nanodiscs with a harsh detergent such as SDS may result in a mixture of denatured monomers and oligomers. Even if successful, such a procedure would not provide additional information beyond what is already established. Our FCS experiments demonstrate that the HIS-tagged protein fraction is homogeneous, containing fluorescently labeled protein and showing a diffusivity consistent with particles the size of nanodiscs. Thus, the protein quality within the nanodiscs is already confirmed.
In some cases, these kinds of synthetic nanodiscs do not successfully embed the protein during the reconstruction process.
Response:
Our evidence strongly suggests otherwise. Membrane proteins typically form aggregates of varying sizes in aqueous solutions. However, our FCS measurements show only one diffusing species, which is too small and homogeneous to represent aggregates. This provides robust evidence that the nanodiscs encapsulate the protein effectively. As pointed out above, it should also be noted that there is no “reconstitution process”, as membrane proteins are encapsulated directly into native nanodiscs, preserving their intact lipid-bilayer environment.
Also, it is not uncommon that the detergent-assisted proteins still can survive in a non-detergent buffer condition especially if they were extracted with a strong detergent such that was used in this study.
Response:
We would like to emphasize that no detergents are used in our protocol. Protein extraction into nanodiscs occurs entirely in a detergent-free system using Glyco-DIBMA. This ensures that the process preserves the protein’s native environment and minimizes artifacts.
As a biochemistry aspect, the authors still do not show direct evidence of whether the resulting sample is properly incorporated into the nanodisc for their following experiment.
Response:
We respectfully disagree. Our FCS measurements directly show that the resulting sample is homogeneous and exhibits a diffusivity corresponding to nanodiscs containing the protein. This confirms proper encapsulation. The reviewer's concern appears to be based on the erroneous assumption that our protocol involves detergent-solubilized proteins, which is not the case.
Comment 3
For the newly added Figure 5 in the revised manuscript, there is no difference between the KvAP WT and the cysteine mutants. Instead, the fluorescence intensity of K102C looks weaker than the WT. This raises a concern that the K102C and E275C mutants should be brighter than the WT because they have additional sites for fluorescence dye conjugation. Also, the authors said that they mutated the native cysteine of KvAP to alanine in lines 259-260. As such, it is unclear how the KvAP WT can be labeled in this condition or if there are still surface-exposed reactive cysteines in the WT. Please describe in the text more clearly.
Response:
We appreciate the reviewer’s observation and would like to clarify that in all KvAP plasmids (wild-type, K102C, E275C), only one cysteine residue per monomer was present. Initially, we tested the labeling of KvAP wild-type, which contains a native cysteine. Subsequently, we mutated this cysteine to alanine and introduced new cysteine residues at positions K102 or E275 to test labeling efficiency further.
We have revised the text for clarity in Line 259:
“To investigate whether only the native cysteine of KvAP is inaccessible to maleimide dye in micelles, it was mutated to alanine, and cysteine residues were introduced at position K102 or E275 (Figure 1A). Structural predictions indicate that these residues are located at the protein surface and should be accessible for maleimide dye binding.”
Comment 4
Regarding the author’s response 8, the authors misunderstood the point of the question. The question was not about their fluorescence intensity, it was about whether there was a possible artifact by this particular chemical species, Glyco-DIBMA. If the altered fluorescence labeling is caused by the use of nanodisc rather than detergent, this should work in other SMA or traditional lipid nanodisc. This is the main scope of this paper, but the authors only tried one chemical species to support this idea.
Response:
We respectfully disagree with the reviewer’s suggestion. There is no evidence to suggest that Glyco-DIBMA itself is fluorescent or contributes to altered fluorescence labeling. Glyco-DIBMA’s glycosylated hydrophilic groups and non-aromatic hydrophobic groups render it chemically inert, as noted in Line 89:
“Moreover, Glyco-DIBMA’s glycosylated hydrophilic groups and non-aromatic hydrophobic groups render it chemically inert.”
Finally, repeating experiments with other polymers is outside the scope of this study for the following reasons:
- Traditional lipid nanodiscs would require the use of detergents, whereas our aim is to develop a detergent-free labeling system.
- Aromatic SMA polymers are more hydrophobic and may interact with the dye, potentially introducing non-specific labeling.
In summary, we are convinced that the experiments suggested by the reviewer would not improve the manuscript.
We hope our responses address the reviewer’s concerns adequately. Please let us know if additional clarifications are required.